# Whole Genome Resequencing Reveals Selection Signals Related to Wool Color in Sheep

**DOI:** 10.3390/ani13203265

**Published:** 2023-10-19

**Authors:** Wentao Zhang, Meilin Jin, Zengkui Lu, Taotao Li, Huihua Wang, Zehu Yuan, Caihong Wei

**Affiliations:** 1State Key Laboratory of Animal Biotech Breeding, Institute of Animal Sciences, Chinese Academy of Agricultural Sciences (CAAS), Beijing 100193, China; m18251871965@163.com (W.Z.); jmlingg@163.com (M.J.); ltt_ltt2020@163.com (T.L.); wanghuihua@caas.cn (H.W.); 2Key Laboratory of Animal Genetics and Breeding on Tibetan Plateau, Ministry of Agriculture and Rural Affairs, Lanzhou Institute of Husbandry and Pharmaceutical Sciences, Chinese Academy of Agricultural Sciences, Lanzhou 730050, China; luzengkui@caas.cn; 3Joint International Research Laboratory of Agriculture and Agri-Product Safety of Ministry of Education of China, Yangzhou University, Yangzhou 225009, China; yuanzehu@yzu.edu.cn

**Keywords:** wool color, whole genome resequencing, Fst, θπ ratio, XP-EHH, selection signal

## Abstract

**Simple Summary:**

The color of wool is an essential trait in sheep which plays a significant role in the textile industry. The color of wool is determined by the presence of various pigments, which can range from white to various shades of brown, gray, black, etc. Understanding the genetics behind wool color is crucial for selective breeding and producing desirable colors for different textile products. By studying the genetic basis of wool color, researchers can identify genes related to pigmentation and develop strategies to enhance or modify wool color. This knowledge contributes to the improvement of wool quality, diversification of textile options, and economic development in the wool industry.

**Abstract:**

Wool color is controlled by a variety of genes. Although the gene regulation of some wool colors has been studied in relative depth, there may still be unknown genetic variants and control genes for some colors or different breeds of wool that need to be identified and recognized by whole genome resequencing. Therefore, we used whole genome resequencing data to compare and analyze sheep populations of different breeds by population differentiation index and nucleotide diversity ratios (Fst and θπ ratio) as well as extended haplotype purity between populations (XP-EHH) to reveal selection signals related to wool coloration in sheep. Screening in the non-white wool color group (G1 vs. G2) yielded 365 candidate genes, among which *PDE4B*, *GMDS*, *GATA1*, *RCOR1*, *MAPK4*, *SLC36A1*, and *PPP3CA* were associated with the formation of non-white wool; an enrichment analysis of the candidate genes yielded 21 significant GO terms and 49 significant KEGG pathways (*p* < 0.05), among which 17 GO terms and 21 KEGG pathways were associated with the formation of non-white wool. Screening in the white wool color group (G2 vs. G1) yielded 214 candidate genes, including *ABCD4*, *VSX2*, *ITCH*, *NNT*, *POLA1*, *IGF1R*, *HOXA10*, and *DAO*, which were associated with the formation of white wool; an enrichment analysis of the candidate genes revealed 9 significant GO-enriched pathways and 19 significant KEGG pathways (*p* < 0.05), including 5 GO terms and 12 KEGG pathways associated with the formation of white wool. In addition to furthering our understanding of wool color genetics, this research is important for breeding purposes.

## 1. Introduction

The influence of wool color on the textile industry dates back a long time [1,2,3], and the colors of wool include white, brown, gray, black, tan, and yellow [2,4,5]. White wool meets the demand for rich colors [1,3,6,7] due to its excellent dyeing ability. Breeding for white wool has always been a priority in sheep farming, due to the high level of pursuit by the textile industry [1,3,5,8,9]. And, with the rise of green concept [3,10,11], natural colored wool is a way to replace traditional printing and dyeing [12]. A comparison of modern and ancient wool products reveals a reduction in the diversity of modern wool colors [5,13], making it imperative to conserve colored wool breeding resources. Research on wool color-related genes can help improve varieties [7,14], increase the economic value and competitiveness of natural wool [14], and provide more options and innovations for textile production of different colored wools [15]. At the same time, understanding wool color characteristics of different genotypes and breeds can help maintain genetic diversity [16], prevent gene loss [16], and promote ecosystem management and conservation [7,17,18,19]. Therefore, continued research on candidate genes for various wool colors remains necessary [14].

Regarding the mining of sheep wool color genes, the initial studies mainly explored the effect of genes on sheep wool color through knockout or mutation of single genes [20]. With the improvement in sequencing technology and the development of in vitro breeding techniques, more and more genes with their variant forms have been discovered and their roles have been determined [9,21]. Currently, technologies such as whole genome sequencing, RNA sequencing, and gene editing techniques are widely used to mine and study wool color genes in sheep [16,17,22,23]. The wool color of sheep is controlled by a series of genes. In the past decades, scientists have successfully identified multiple genes related to wool color in sheep. The *MC1R*, *ASIP*, *TYRP1*, *KIT* and *MITF* loci are important in biology and genetics, and they play key roles in biological processes such as formation and distribution of coat color, pigment production and distribution, and cell migration. The study of these genes not only contributes to our in-depth understanding of the genetic mechanism of coat color but also provides a basis for the improvement and selection of sheep breeds with specific coat color characteristics. The study of key genes for coat color not only is important for animal husbandry but also provides valuable information for biological and medical research, as well as deepening the understanding of ecology and evolution. Under the action of the *MC1R* gene, melanocytes produce melanin and deposit it into the hair follicle, which results in a black or brown coat color in sheep [24,25]. In contrast, under the action of the *ASIP* gene, melanin production and deposition is inhibited, leading to light pigmentation [24]. The ratio of the expression of these two genes allows the wool color to be presented in the black to reddish-brown range [24]. TYR is a key enzyme in the regulation of melanogenesis, and mutations in *TYR* lead to the production of white wool [24,26,27]. TYRP1 is an important enzyme in the synthesis of true melanin [24,26,27]. Mutations in *TYRP1* result in the inability to convert the brownish 5,6-dihydroxyindol into the blackish eumelanin, which affects the shade of brown color of wool [24,28]. *KIT* [19], *MLPH* [24], and *KIF5A* [29] are commonly recognized as genes that mediate the formation and distribution of pigment granules in melanocytes which are associated with the translocation of melanosomes. In contrast, the *PMEL* [27] gene is involved in melanosome structure, and its variation can inhibit melanosome formation, resulting in melanin dilution [24,25]. In addition, the regulatory mechanisms of transcription factors are also closely related to sheep wool color. Transcription factors are a class of proteins that can bind to gene DNA and regulate gene expression, such as SOX10 [24] and MITF [30], which have been shown to be essential for melanocyte differentiation and maturation. *MITF* is a key regulator of pigmentation, and variations in *MITF* have been associated with the formation of light-colored wool [24,30]. Variations in *IRF4* lead to lighter coloration [31,32]. Mutations in *DCT* result in increased production of eumelanin and decreased production of pheomelanin in melanocytes [25,27,33,34]. The *MSG1* (*CITED1*) gene enhances melanin production in B16 cells [35]. In previous studies, black and white wools were mainly used, followed by brown and tan, while other colors of wools were less studied.

Wool color is complex and affected by the interaction of multiple genes [5,24], and although some relevant genes have been identified, research is still ongoing [14]. Different breeds of sheep may be caused by different genes even if they have the same color wool [19,24], and they have different genetic variants [7,19], requiring in-depth study of the mechanism of wool color [24]. In addition, gene interactions and environmental factors also affect wool color [19], which is an issue that may require further research in the future.

We studied the coat color of different breeds of sheep by whole genome resequencing to obtain more comprehensive and detailed data [9,19,21]. Employing three signal analysis methods [36] (population differentiation index (Fst [36]), nucleotide diversity ratio (θπ ratio [37,38,39]), and cross-population extended haplotype homozygosity (XP-EHH [40])) allowed us to obtain comprehensive insights into selection signals, resulting in improved reliability and understanding of the evolution and adaptations of sheep breeds, as well as the effects of natural and artificial selection.

## 2. Materials and Methods

### 2.1. Ethics Statement

All experimental work on sheep was approved by the Animal Ethics Committee of the Institute of Animal Science, China Academy of Agricultural Science (protocol code IAS 2022-7 and 25 February 2022).

### 2.2. Sample Collection and Sequencing

Jugular vein bloods were collected from fifteen sheep breeds (Table 1) in 2019, including Bashbay sheep (BAS), Duolang sheep (DUL), Altay sheep (ALT), Qira Black sheep (QIB), Turfan Black sheep (TUB), Guide Black Fur sheep (GBF), Ninglang Black sheep (NLB), Shiping Gray sheep (SPG), German Mutton Merino (GME), Poll Dorset (DOP), Large-tailed Han sheep (LTH), Guangling large-tailed sheep (GLT), Hu Sheep (HUS), Tong Sheep (TON), and Lanzhou Large-tailed sheep (LLT). DNA extraction and library construction were then performed. Next, Illumina PE150 was used to sequence the sheep, and the resequenced data were used for further analysis.

We carefully selected healthy, breed-typical individuals at the age of 1 year to ensure a representative and comparable cohort of animals. Whenever possible, we utilized full/half siblings to minimize individual variations. The farm provided high-quality feed and ensured access to clean water. Moreover, it offered a suitable and comfortable feeding environment with appropriate space allocation, dry bedding, and a controlled temperature range. Regular health checks and vaccination programs were implemented along with strict hygiene practices. Reasonable exercise opportunities were provided while ensuring adequate rest for the animals. Additionally, effective breeding management was carried out alongside regular inspection and maintenance of equipment.

### 2.3. Alignments and Quality Control

The raw reads of fastq format were first processed through a series of quality control procedures using FastQC to ensure reliable reads in the subsequent analyses. The standards of quality control were followed, including (1) removing reads with ≥10% unidentified nucleotides (N); (2) removing reads with >20% bases having phred quality less than 5; (3) removing reads with >10 nt aligned to the adapter, allowing ≤ 10% mismatches; and (4) removing putative PCR duplicates generated by PCR amplification in the library construction process (reads 1 and 2 of 2 paired-end reads that were completely identical).

Valid high-quality sequencing data were aligned to the reference genome (GCF_016772045.1_ARS-UI_Ramb_v2.0) using BWA software (v 0.7.17) [41] with the following parameters: mem -t 4 -k 32 -M. The resulting alignments were processed using SAMTOOLS [42] to remove duplicates, employing the parameter rmdup. In order to enhance the accuracy of data analysis, high-quality SNPs [43] meeting the following criteria were selected: (1) SNPs with a depth of coverage greater than 2; (2) SNPs with a proportion of MIS (deletions) less than 10%; (3) SNPs with a minimum allele frequency (MAF) greater than 5% [44].

### 2.4. Population Structure Analysis

Before conducting the analysis, all single nucleotide polymorphisms (SNPs) underwent trimming using the indep-pairwise [45] function of PLINK 1.09 software [46]. The trimming process involved applying specific parameters, including a non-overlapping window of 25 SNPs, a step size of 5 SNPs, and a threshold of 0.05 for r2, in order to obtain a set of independent SNP markers. To examine the clustering patterns within the population, we conducted a principal component analysis (PCA) using PLINK 1.09 [46]. Additionally, to assess the genetic relatedness among individuals, we constructed neighbor-joining (N-J) trees [47] using MEGA (v 7.0) software [48] and visualized them using ITOL (v 6) software [49] (https://itol.embl.de/upload.cgi, (accessed on 9 June 2023)). Furthermore, to evaluate the extent of population stratification and to validate the findings from PCA and N-J trees, we employed ADMIXTURE (v 1.3) software [50] to construct the population genetic structure, with k values ranging from 2 to 9.

### 2.5. Analysis of Selection Signals

Resequenced data are rich in variation information yet are fraught with noise and false alarms. By first using Fst [36] and θπ ratio [37,38,39] screening, an initial set of candidate selection signals can be quickly obtained, reducing the time and resources required for subsequent analysis. The true selection signal can be more accurately identified and potential false positives can be eliminated by subsequent verification and validation using XP-EHH [40]. As a starting point for analysis, BAS, DUL, ALT, QIB, TUB, GBF, NLB, and SPG were categorized into G1 and GME, DOP, LTH, GLT, HUS, TON, and LLT were categorized into G2. The Fst and θπ values were computed by employing VCFtools (version 0.1.15) software [51]. The analysis incorporated specific parameters: -fst-window-size 50,000 and -fst-window-step 50,000. Subsequently, the obtained values were utilized to derive the θπ ratio [37,38,39]. The selected genomic intervals of the G1 and G2 populations associated with wool color traits were screened by comparing the Fst and pi values of the G1 and G2 populations. Then, we use population marker information to estimate the haplotype of each chromosome by fastphase 1.4 [52] with the options set to −Ku40 −Kl10 −Ki10. XP-EHH scores were calculated using haplotype information from the XP-EHH program at http://hgdp.uchicago.edu/Software/, (accessed on 9 June 2023) to determine whether selection had occurred in the experimental (G1 or G2) population [40]. Using the sliding window method, XP-EHH values were then calculated with a window size of 50 kb and a step size of 20 kb. Next, the mean was computed for each SNP in the sliding window. A negative XP-EHH score means that selection has taken place in the reference population, in contrast to a positive XP-EHH score, which represents that selection has occurred in the experimental population. 

The problem of small sample sizes in varieties can be addressed to some extent by combining multiple methods for analysis. By combining multiple methods, the ability to detect genetic variation can be improved and the reliance on large sample sizes can be reduced [40,53,54]. Each method has its own unique information and limitations, so joint analysis can combine the strengths of different methods and increase the sensitivity and accuracy of detection of genetic signals [55,56,57]. In addition to increasing detection power, the combined analysis of three methods can be independently validated, providing complementary information [40,53,58,59]. Consistent results from multiple methods can increase confidence in the signal. When multiple methods indicate the presence of the same genetic variant signal, the authenticity of the signal can be confirmed with greater confidence. Different methods have different characteristics and preferences for detecting genetic variation. By utilizing multiple methods in combination, a comprehensive understanding of the characteristics and patterns of genetic variation can be obtained from different perspectives, better revealing the potential biological significance. 

### 2.6. Detection and Annotation of Candidate Genes

Different breeds of wool colors have different heritabilities, mostly low to medium heritabilities, but some have high heritability [60,61,62,63,64]. It is possible that genes associated with wool color are located in regions with low genetic diversity. Therefore, in this study, the threshold for selection signal analysis has been adjusted to the top 5% to avoid overlooking candidate genes that may be involved in wool color [65]. Then, the loci within the window where Fst [36], θπ ratio [37,38], and XP-EHH [40] were top5% were extracted as significant SNP loci, namely the candidate loci for the selection signal. Areas 50 kb upstream and downstream of the candidate loci were regarded as selection signaling regions. We used ANNOVAR software (https://annovar.openbioinformatics.org/en/latest/, accessed on 9 June 2023) [66] to annotate the genes with the sheep reference genome. Finally, the Venn diagram was created on the basis of the candidate genes derived from Fst, θπ ratio, and XP-EHH.

### 2.7. Candidate Gene Enrichment Analysis

To uncover the function and the mechanism of expression regulation of the genes, a functional enrichment analysis was performed. Candidate gene functional enrichment was performed using DAVID 6.8 [67] (https://david.ncifcrf.gov/, (accessed on 15 June 2023)), with gene symbol as the input parameter and *Ovis_aries* selected as the background organism. We tallied the number of genes that were enriched in these GO [68] terms and evaluated the significance of their enrichment by means of the hypergeometric distribution test. These genes were analyzed for KEGG [69] enrichment by means of Kobas 3.0 [70] (http://kobas.cbi.pku.edu.cn/kobas3/genelist/, (accessed on 15 June 2023)), with *Ovis_aries* selected for background organism, using the Hypergeometric test/Fisher’s exact test as the statistical method. The terms and pathways with *p*-value < 0.05 were judged to be significant.

## 3. Results

### 3.1. Genetic Variation and Population Genetic Analysis

Whole genome resequencing with an average coverage of 7.6× was performed on 48 sheep individuals in this study. A total of 9,581,315,830 reads were obtained after alignment to the sheep reference genome (ARS-UI_Ramb_v2.0), covering 98.03% of the reference sequence. A coverage of 98.03% signifies that the likelihood of missing important information or encountering errors is minimized, thereby ensuring more comprehensive and accurate genomic insights. Furthermore, high coverage facilitates enhanced confidence and interpretability, particularly when examining variant annotations, identifying mutation sites, and exploring genomic structure and function. Consequently, this dataset proves valuable for subsequent studies involving population structure analysis and identification of selection signals. After variant calling and quality control, a total of 22,133,207 SNPs were identified. Statistical results of SNPs showed that variants mainly occurred in intergenic interval, followed by intronic interval, exonic interval, etc. Among the exonic variants, there were 82,379 non-synonymous SNPs and 68,110 synonymous SNPs (Table 2). The TS/TV ratio was determined to be 1.9, closely approximating 2. This observation implies a relatively balanced distribution of SNPs across the population, indicative of a normalized genomic population structure. These findings establish a solid foundation of reliable data for further investigations into population structure and the identification of potential selection signals. 

Firstly, a map of the worldwide distribution of sheep breeds was created (Appendix A). Then PCA, phylogenetic tree construction, and population structure analysis were executed on the fifteen sheep populations using the received SNP datasets to understand the genetic relationships and differences between different wool color sheep populations from a genome-wide perspective. According to the PCA results (Figure 1a), PC1 and PC2 explained 5.33% and 4.04% of the genetic variation, respectively; the 15 breeds clustered into two groups, European sheep breeds and East Asian sheep breeds; Yunnan sheep breeds were significantly separated from other East Asian sheep breeds; and Tibetan sheep breeds were slightly separated from Kazakh and Mongolian sheep breeds. The non-white wool sheep breeds (GBF, SPG, and NLB) and white wool sheep breeds (DOP and GME) were separated from the population via PC1 and PC2 (Appendix A). The non-white wool sheep breeds (BAS, DUL, ALT, QIB, and TUB) could be separated from the remaining sheep population following PC3 (Appendix A). The population genetic structure (Figure 1b,d) was constructed using ADMIXTURE software to confirm the accuracy of the results obtained from PCA. With K = 2, the blue background was dominant, and there was a clear transition from European sheep breeds to East Asian Kazakh, Tibetan, and Mongolian sheep breeds to Yunnan sheep breeds; Yunnan sheep breeds and European sheep breeds were clearly separated from other breeds; and when K = 3, Tibetan sheep (GBF) were separated from other breeds. The population genetical structure results confirmed the results of PCA. The results of the N-J tree (Figure 1c) are somewhat different from those of PCA and STRUCTURE. The PCA results are the same as those of STRUCTURE, which suggests that the position of individuals in genetic space is consistent with their genetic components among different genetic groups. The N-J tree is inconsistent with the PCA and STRUCTURE results, which suggests that there are differences in the phylogenetic relationships between species and that further study and consideration of other possible factors and explanations are needed.

### 3.2. Analysis of Selection Signals

Within the non-white wool group (G1 vs. G2), 3944 top 5% selection signals were screened by the joint Fst&θπ ratio (Figure 2a); 8223 top 5% selection signals were screened by XP-EHH (Figure 2b). Upon ANNOVAR annotation of the screened candidate SNPs, 544 and 1061 candidate genes associated with colored wool color were detected, respectively. After constructing the Venn diagram, 365 overlapping candidate genes were obtained (Figure 2d), 2431 and 4250 selection signals in the top 5% of Fst&θπ ratio (Figure 2a) and XP-EHH (Figure 2c) were screened in the white wool group (G2 vs. G1), and 388 and 625 candidate genes were derived by annotation. Ultimately, 214 overlapping candidate genes were screened using the Venn diagram [71] (Figure 2e).

### 3.3. Enrichment Analysis

Both GO [68] and KEGG [69] enrichment were performed on candidate genes screened in the sheep genome using Fst [36], θπ ratio [37,38,39], and XP-EHH [40]. Initially, there are 21 significant GO terms (Appendix A) in the non-white wool group (G1 vs. G2), including 3 noteworthy biological processes (BP), 10 noteworthy cellular components (CC), and 8 noteworthy molecular functions (MF) (*p*-value < 0.05, Figure 3a). Inquiring about the role of GO terms (http://geneontology.org/, (accessed on 29 June 2023)) and the results of previous studies, the following 17 GO terms are associated with non-white wool formation: myosin II complex (GO:0016460, *MRCL3*, *LOC101105123*, *MYH10*), protein kinase binding (GO:0019901, *PPP1CB*, *ARHGAP33*, *TRAF3*, *CHEK2*, *SPDYA*, *CCNYL1*, *NR3C1*, *KIZ*, *CDC25A*, *MAPK4*), actin monomer binding (GO:0003785, *NOS3*, *PRKCE*, *MTSS1*), etc. (Appendix A). In contrast, there were nine significant GO terms (Appendix A) in the white wool group (G2 vs. G1), including three noteworthy BPs, five noteworthy CCs, and one noteworthy MF (*p*-value < 0.05, Figure 3c). The following 5 GO terms are associated with white wool formation according to the role of GO terms (http://geneontology.org/, (accessed on 29 June 2023)) and the results of previous studies: myelination (GO:0042552, *SLC8A3*, *ATRN*, *ACER3*), protein kinase complex (GO:1902911, *NEK10*, *IGF1R*), RNA polymerase II transcription factor activity, sequence-specific DNA binding (GO:0000981, *ISL2*, *HOXA3*, *EVX1*, *HOXA7*, *ZFHX4*, *HOXA6*, *HOXA5*), etc (Appendix A).

Then, in the non-white wool group (G1 vs. G2), 49 significant KEGG enrichment pathways (Appendix A) were identified (*p*-value < 0.05, Figure 3b). From the analysis of the role of KEGG (https://www.kegg.jp/kegg/pathway.html, (accessed on 29 June 2023)) and the results of previous studies, the following 27 KEGG pathways are associated with non-white wool formation: arginine and proline metabolism (oas00330, *MAOA*, *NOS3, AGMAT*), purine metabolism (oas00230, *PDE4B*, *ENTPD5*, *PAPSS2*, *NT5E*, *PDE11A*), retrograde endocannabinoid signaling (oas04723, *NDUFS1*, *GABRR1*, *MAPK10*, *GABRA1*, *GRM5*), etc. (Appendix A). There were 19 significant KEGG enrichment pathways (Appendix A) in the white wool group (G2 vs. G1) (*p* < 0.05, Figure 3d). When looking at the role of KEGG (https://www.kegg.jp/kegg/pathway.html, (accessed on 29 June 2023)) and the results of previous research, the following 12 KEGG pathways are related to the development of white wool formation: valine, leucine, and isoleucine degradation (oas00280, *PCCB*, *HIBADH*, *ACAT2*, *ALDH6A1*); lysine degradation (oas00310, *COLGALT2*, *NSD3*, *ACAT2*); pyruvate metabolism (oas00620, *ACSS2*, *ACAT2*); etc. (Appendix A).

Based on the pathways that were screened for association with wool color (Figure 3e,f), we constructed Sankey diagrams to predict genes involved in the formation of non-white and white wool (Table 3).

## 4. Discussion

### 4.1. Sample Control and Population Genetic Analysis

In this study, we performed whole genome resequencing on 48 sheep samples. Individual variations may also potentially affect the analysis of selection signals. To mitigate these differences, this research focuses on selecting representative samples to minimize individual disparities. Additionally, a joint analysis method is employed to address the limitations posed by the small sample size, compensating for individual variations and reducing potential errors. This approach aims to improve the screening efficiency and reliability of selection signals by covering a broader and more comprehensive genomic region using the three combined methods [40,53,54].

According to the PCA and ADMIXTURE results, it was found that there were high genetic similarities and consistent genetic components among the four populations of European, Yunnan, Kazakh, and Mongolian sheep, which were consistent with their breeding history. The Tibetan sheep (GBF) is very close to the Kazakh and Mongolian sheep in PCA and has the genetic components of Yunnan sheep in the population structure analysis; GBF is a local breed, which has been selected and bred for a long time, and according to its geographic location, it is assumed that there is a genetic exchange with Yunnan sheep, Kazakh, and Mongolian sheep. The QIB in the N-J tree is very inconsistent with PCA and ADMIXTURE. According to the investigation, QIB was bred in the late 19th century by merchants and pilgrims who brought back lambskin sheep and other black lambskin sheep from overseas and crossed them with local ewes. The breeding time is relatively short, so it is closer to the European sheep in terms of phylogenetic relationship.

### 4.2. Selective Signal Analysis

Wool color formation is a complex process regulated by a variety of factors and mechanisms, mainly involving the development of pigment cells [72,73], pigment synthesis [72,73], pigment transport and release [72,73,74,75], pigment particle distribution [24,74], and other processes. The overall appearance of wool color in different animals also depends on the distribution pattern of pigment granules. Due to the specific origins of melanoblasts in certain regions of the neural crest during embryonic development, there are specific time frames within which migrating cells must reach their designated positions in the skin [72,73]. Failure to do so can lead to areas of the skin lacking pigment cells, resulting in patches of white coloration known as leucism [24]. Impaired melanogenesis ultimately leads to a complete absence of pigment. This phenomenon is most commonly observed on the legs, abdomen, and forehead since these areas are farthest from where melanoblasts originate and therefore require more time for cell migration [19]. Overall, the formation of wool color is regulated by gene expression, cell–cell interactions, and hormonal regulation [74].

#### 4.2.1. GO Terms and Pathways Associated with Non-White Wool

Based on the enrichment results, we reviewed previous literature and found that the following GO [68] terms and KEGG [69] pathways are associated with processes such as pigment cell development, pigment synthesis, pigment transport and release, and distribution of pigment granules. Phosphoric diester hydrolase activity (GO:0008081) regulates the levels of intracellular second messenger molecules, such as intracellular calcium ions (calcium signaling pathway (oas04020)) [76,77,78], cAMP [79], and cGMP (cGMP-PKG signaling pathway (oas04022)), which are important in the regulation of pigment aggregation/dispersal and production [76,80] through protein kinase binding (GO:0019901). Deletion of the ATP-binding region in the structural domain of the kinase inhibits pigment dispersion, and thus, ATP binding (GO:0005524) plays a key role in pigment transporter and plays an important role in energy provision [76]. Thus, protein kinase binding (GO:0019901) and ATP binding (GO:0005524) are closely interrelated during pigment transport and jointly regulate intracellular signaling pathways and the dynamic distribution of pigment granules. Actin filaments and myosin motors are required for vesicle transport and retention of organelles in specific locations [81,82,83,84]. Actin filament-monomer turnover leads to the aggregation and dispersion of pigments [82], and thus, actin monomer binding (GO:0003785) and the regulation of actin cytoskeleton (oas04810) maintains a rational distribution of intracellular pigments. Myosins are categorized into (muscle) myosins and non-muscle myosins [85]. Actin-dependent myosin II, known as myosin II complex (GO:0016460), drives pigment granule aggregation [86,87]. The actin–myosin system plays a role in particle transport of melanin carriers in rats [88], fish [89], and amphibians [90]. In contrast, non-muscle myosin II isoforms may play a role in pigment aggregation in crustacean and vertebrate pigment cells [82,84,85,91]. Focal adhesion assembly (GO:0048041) is a specialized structure that connects the cell cytoskeleton to the extracellular matrix [92]; focal adhesion (oas04510) plays a key role in the interaction between the extracellular matrix and pigment cells [92], which affects the development and distribution of pigment cells [93]. TERMS and PATHWAYS associated with neurotransmitters and their receptors may affect pigment synthesis and release [94], excitatory postsynaptic potential (GO:0060079) [94], glutamatergic synapse (GO:0098978) [94,95], dendritic spine (GO:0043197) [77,96,97], axon (GO:0030424) [98], postsynaptic membrane (GO:0045211) [99,100], integral component of postsynaptic density membrane (GO:0099061), axon guidance (oas04360), dopaminergic synapse (oas04728) [94,101], neuroactive ligand–receptor interaction (oas04080) [23], glutamatergic synapse (oas04724) [94,95], and long-term potentiation (oas04720) [100]. The histone deacetylase complex (GO:0000118) plays a key role in melanocyte development [102]. The transcriptional repressor complex (GO:0017053) affects hair color phenotype by repressing the transcription of genes, thereby reducing or preventing the expression of specific genes [103,104,105,106]. The chloride channel complex (GO:0034707) may regulate melanin synthesis by modulating melanosome pH [78,107]. The cysteine-type endopeptidase activity (GO:0004197) may be involved in regulating the degradation or activation of key enzymes in the pigment synthesis process, affecting the production and regulation of pigmentation [108]. Zinc ion binding (GO:0008270) contributes to the binding of zinc ions to TYR and TYRP1, promoting melanogenesis [109,110]. The MAPK signaling pathway (oas04010) [111,112,113] and Ras signaling pathway (oas04014) [114,115,116] have important roles in the proliferation and differentiation of melanocytes, and the MAPK signaling pathway is directly linked to the synthesis of wool color [112]. The Wnt signaling pathway (oas04310) is an important pathway in melanin synthesis [112], pigment aggregation [117], and melanocyte stem cell differentiation [117,118,119]. The PI3K-Akt signaling pathway (oas04151) is critical for melanocyte proliferation and apoptosis [120,121,122]. Endocytosis (oas04144) mediates melanin transfer between melanocytes and keratinocytes [123]. Huntington’s disease (oas05016) affects the release of α-melanocyte-stimulating hormone (α-MSH) [124], which in turn affects melanin synthesis and release [125]. The oxytocin signaling pathway (oas04921) regulates the production and release of oxytocin, and oligopeptides such as oxytocin stimulate melanin production [126]. Nicotine addiction (oas05033) affects melanin synthesis [127,128,129]. The Rap1 signaling pathway (oas04015) promotes melanocyte proliferation by activating the downstream protein kinase C-Raf and ERK signaling pathways [113,116,130,131], in addition to regulating pigment synthesis [132] as well as pigment cell migration and adhesion [113]. Downregulation of tight junction (oas04530)-related gene expression results in reduced cell–cell junctions that contribute to melanosome/melanin transfer [133]. Vascular smooth muscle contraction (oas04270) may be associated with melanin synthesis [134]. The IL-17 signaling pathway (oas04657) regulates the production and action of IL-17, which can inhibit melanin production [135]. Purine metabolism (oas00230) may be associated with golden pigment production [136,137,138], which correlates with the blue phenotype [139], and may also affect melanin synthesis [140,141,142]. Glycine, serine, and threonine metabolism (oas00260) also correlate with the blue phenotype [139,143]. The histidine metabolism (oas00340) may be associated with melanin deposition [144,145]. The phosphatidylinositol signaling system (oas04070) controls melanocyte proliferation and differentiation [146]. The VEGF signaling pathway (oas04370) affects melanin synthesis [147]. Carotenoids produce yellow, orange, and red colors, and retrograde endocannabinoid signaling (oas04723) may be associated with carotenoid-based coloration [148].

#### 4.2.2. GO terms and Pathways Associated with White Wool

Arginine and proline metabolism (oas00330) affects and regulates the function of pigment cells, which leads to the lightening or whitening of the animal’s wool color [149]. Myelin and melanocytes share common progenitors, and thus, myelination (GO:0042552) may affect melanocyte formation, which in turn leads to leucism [72,73]. The structure and modification of chromatin (GO:0000785) plays an important role in the regulation of gene expression and phenotypic features, and it may be involved in the synthesis and formation of skin and hair pigmentation [150,151,152,153]. The protein kinase complex (GO:1902911) plays an important role in cell signaling and regulation and is associated with pigment cell differentiation and pigment synthesis [30,154,155]. The mitochondrion (GO:0005739) is involved in many cellular functions, including energy production and intracellular signaling, and its dysfunction is associated with hypopigmentation [156,157]. RNA polymerase II transcription factor activity and sequence-specific DNA binding (GO:0000981) are associated with melanocyte development and differentiation [158]. Valine, leucine, and isoleucine degradation (oas00280) is associated with hair color, and complexes of branched-chain amino acids which are potential depigmenting agents inhibit melanin synthesis [159,160]. Carbon metabolism (oas01200) provides the carbon framework for melanin synthesis, but low carbon conditions are not favorable for melanin synthesis [161]. Terpenoid backbone biosynthesis (oas00900) is associated with the synthesis of many natural products, some of which (carotenoids) may be related to coat color, skin color, and melanin [162,163,164]. Selenocompound metabolism (oas00450) may be related to melanin because selenium is a trace element that has an effect on melanin synthesis [165,166]. Lysine degradation (oas00310) prevents melanin pigment formation by inhibiting tyrosinase activity, which in turn leads to depigmentation [167]. Nicotinate and nicotinamide metabolism (oas00760) decreases melanin synthesis [168]. Pyruvate metabolism (oas00620) inhibits melanin biogenesis [169]. Glycerophospholipid metabolism (oas00564) and purine metabolism (oas00230) are associated with melanin deficiency [170]. The dysregulation of DNA replication (oas03030) may induce melanocyte decay [171]. Transcriptional misregulation in cancer (oas05202) is involved in the regulation of melanin deposition [172]. Glycine, serine, and threonine metabolism (oas00260) may also play a role in the white phenotype [139,143].

#### 4.2.3. Genes Associated with Wool Color

Based on the significantly different genes obtained using the multiple selection signal analysis method and the cross-pathway genes in the Sankey diagram of wool color-related pathways, we have obtained some genes reported to be related to wool color formation after checking the related research literature to confirm previous studies and to better reflect the accuracy of this paper. However, we must point out that genetic differences between breeds may also affect the association between genes and coat color, since the two sets of samples in this study cover different breeds. In addition to this, candidate genes may not produce consistent effects in coat color phenotypes across breeds. *PDE4B* is associated with melanin deposition [173]. Nie et al. found that *GMDS* may be associated with skin color regulation from a genome-wide association analysis [174]. GATA1 and RCOR1 are major transcription factors for melanin formation [175], and *GATA1* affects the formation of the red phenotype [176,177]. The *MAPK4*-mediated MAPK4/MAPK6 pathway affects melanocyte proliferation and differentiation [113]. *TECRL* is associated with hyperpigmentation of the head [112]. *MAPK10* is an important gene involved in melanin synthesis [112]. *SLC36A2* may be associated with melanogenesis [178], and it also is a candidate gene for cream, pearl, and champagne dilution phenotypes in horsehair [179,180]. Mutations in exon 2 of the *SLC36A1* gene are a key locus responsible for the champagne coat color in horses [179]. *MAOA* is associated with the leucism phenotype [94,181]. *NOS3* may theoretically reduce melanogenesis [182]. *PDE4B* is involved in melanin synthesis [183,184] and were exclusively associated with tanning ability [184]. The overexpression of *GABRR1* inhibits melanin stem cell regeneration [185]. The *GRM5* gene is associated with skin and hair pigmentation [186,187]. *PPP3CA* may be associated with color variation [188]. *PPP1CB* was shown to regulate actin filament polymerization and/or reorganization [189], regulating the distribution of pigment granules. Mutations in *ABCD4* cause hyperpigmentation of the skin, leading to lighter hair color [190,191]. *VSX2* affects human skin pigmentation [192] and is also a candidate gene for vertebrate retinal pigmentation [193,194]. The *ITCH* gene regulates the expression of the *ASIP* gene, which leads to the white wool phenotype [195]. *NNT* can inhibit melanogenesis by suppressing *MITF* gene expression [196]. However, another study showed that *NNT* can inhibit redox-dependent hyperpigmentation by a mechanism independent of UVB and *MITF* [197,198]. *POLA1* has been associated with impaired pigmentation [199]. *PDE3A* regulates the production of cAMP and cGMP, which in turn may affect pigment aggregation and dispersion through a number of termes and pathways [200,201]. *MCM6* is associated with pigmentation around the eyes of cattle [202] and may be related to the formation of orange and blue skin color in lizards [138]. In addition to this, *MCM6* influences the normal development of melanocytes through DNA replication [203,204,205]. *POLA1* has been shown to be associated with reticulate skin hyperpigmentation in humans [206,207] and may be related to hyperpigmentation in threespine stickleback [206]. *IGF1R* belongs to a family of tyrosine kinase receptors on cell membranes, and its aberrant expression hinders melanocyte pigmentation, proliferation, and migration [208,209,210]. *HMGA2* can greatly strength melanocyte stem cell activation and translocation [211,212]. *MEIS1* leads to the formation of ectopic pigment cell clumps [213]. *HOXA10* gene upregulates the *DKK1* gene to regulate skin pigmentation [214,215] and may influence the black and white hair follicle phenotype in goats [215]. Abnormal function of *DAO* enzymes may be associated with a number of skin pigmentation disorders [216,217,218].

Based on the screened pathways associated with wool color, we constructed Sankey diagrams and queried the cross-pathway genes. For some cross-pathway genes that were unknown to be associated with wool color traits, it was hypothesized that *CALML4*, *GRIN1*, *MYLK*, *FGF18*, and *FGFR2* were associated with non-white wool formation and that *ACAT2*, *PCCB*, *ALDH6A1*, and *ACSS2* were associated with white wool formation.

## 5. Conclusions

We have identified a range of genes that play pivotal roles in the formation and regulation of wool color. Among them, *PDE4B*, *GMDS*, *RCOR1*, *TECRL*, *MAPK10*, *SLC36A2*, *SLC36A1*, *MAOA*, *GABRR1*, *GRM5*, *PPP3CA*, and *PPP1CB* were associated with melanin stem cell regeneration, melanocyte proliferation and differentiation, melanin synthesis and distribution, as well as color variation, affecting the formation of the non-white wool phenotype; *ABCD4*, *VSX2*, *ITCH*, *NNT*, *POLA1*, *PDE3A*, *MCM6*, *POLA1*, *IGF1R*, *HMGA2*, *MEIS1*, *HOXA10*, and *DAO* were involved in impediments to melanocyte pigmentation, proliferation, and migration, which influence the formation of the white wool phenotype. In addition, we found that some genes (*CALML4*, *GRIN1*, *MYLK*, *FGF18*, *FGF2*, *ACAT2*, *PCCB*, *ALDH6A1*, and *ACSS2*) may be involved in wool color formation, which needs to be further verified. Our results will help to better promote sheep wool improvement breeding, which is crucial for the development of the white wool industry in China.

## Figures and Tables

**Figure 1 animals-13-03265-f001:**
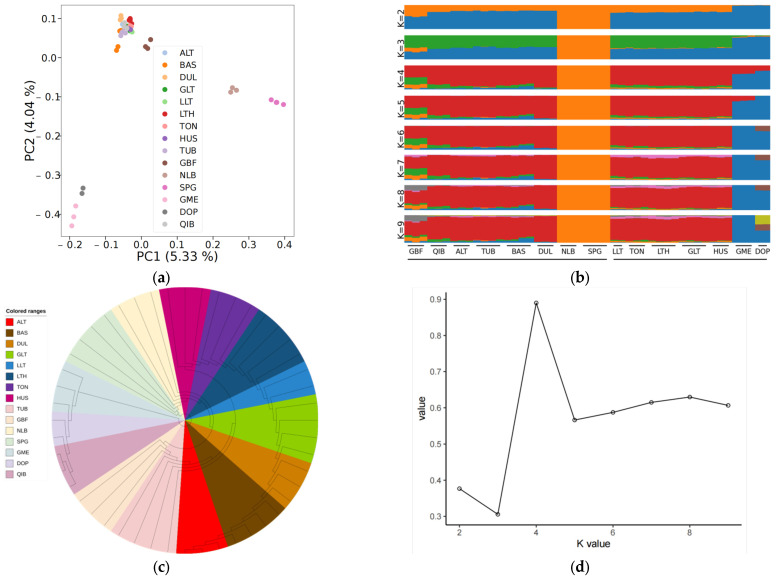
World distribution map of sheep breeds and population genetic structure analysis: (**a**) principal component analysis (PCA); (**b**) population structure analysis (Different colors represent different components of ancestry); (**c**) neighbor-joining tree; (**d**) cross-validation error.

**Figure 2 animals-13-03265-f002:**
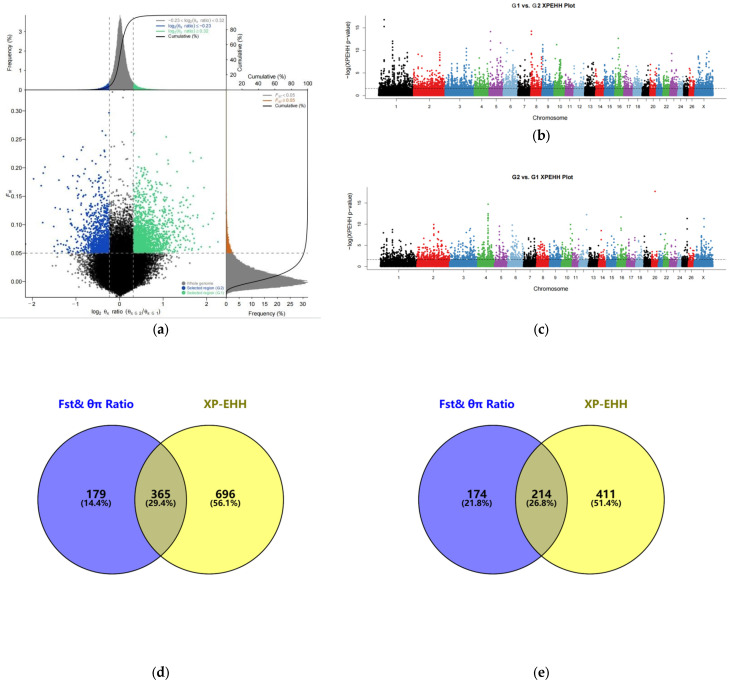
Selection signal analysis: (**a**) Fst and θπ ratio selection elimination analysis plots; (**b**) genome-wide distribution of XP-EHH (G1 vs. G2) (Different colors represent different chromosomes); (**c**) genome-wide distribution of XP-EHH (G2 vs. G1) (Different colors represent different chromosomes); (**d**) Fst, θπ ratio, and XP-EHH screened for overlapping genes (G1 vs. G2); (**e**) Fst, θπ ratio, and XP-EHH screened for overlapping genes (G2 vs. G1).

**Figure 3 animals-13-03265-f003:**
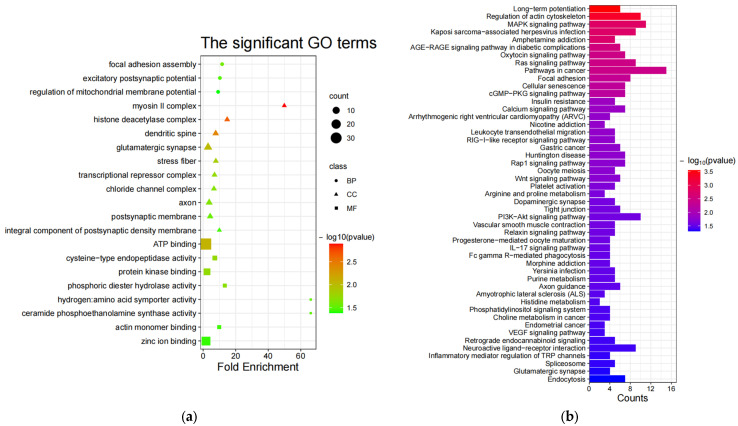
GO enrichment and KEGG enrichment results. (**a**) The most enriched GO terms (G1 vs. G2); (**b**) KEGG pathway enrichment (G1 vs. G2); (**c**) the most enriched GO terms (G2 vs. G1); (**d**) KEGG pathway enrichment (G2 vs. G1); (**e**) Sankey diagrams for relevant pathways (G1 vs. G2); (**f**) Sankey diagrams for relevant pathways (G2 vs. G1).

**Table 1 animals-13-03265-t001:** Information on the sheep populations in this study.

NO.	Breed	Abbr.	Photo	Category	Size	Color
1	Bashbay sheep	BAS	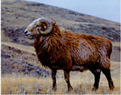	Domestic_East Asia_Kazakh	4	Brown wool with white face
2						
3						
4						
5	Duolang sheep	DUL	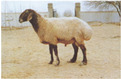	Domestic_East Asia_Kazakh	3	Gray white wool with dark gray head and limbs, tawny neck
6						
7						
8	Altay sheep	ALT	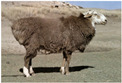	Domestic_East Asia_Kazakh	3	Brown red wool with white head
9						
10						
11	Qira Black sheep	QIB	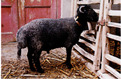	Domestic_East Asia_Kazakh	3	Black brown wool
12						
13						
14	Turfan Black sheep	TUB	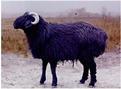	Domestic_East Asia_Kazakh	4	Black wool
15						
16						
17						
18	Guide Black Fur sheep	GBF	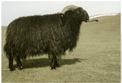	Domestic_East Asia_Tibet	3	Black red wool
19						
20						
21	Ninglang Black sheep	NLB	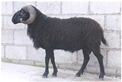	Domestic_East Asia_Yunnan	3	Black wool
22						
23						
24	Shiping Gray sheep	SPG	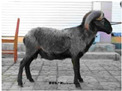	Domestic_East Asia_Yunnan	4	Cyan wool with black limbs
25						
26						
27						
28	German Mutton Merino	GME	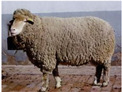	Domestic_Europe	3	White wool
29						
30						
31	Poll Dorset	DOP	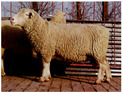	Domestic_Europe	2	White wool
32						
33	Large-tailed Han sheep	LTH	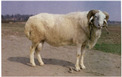	Domestic_East Asia_Mongolia	4	White wool
34						
35						
36						
37	Guangling large-tailed sheep	GLT	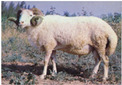	Domestic_East Asia_Mongolia	4	White wool
38						
39						
40						
41	Hu Sheep	HUS	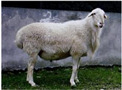	Domestic_East Asia_Mongolia	3	White wool
42						
43						
44	Tong Sheep	TON	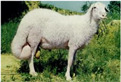	Domestic_East Asia_Mongolia	3	White wool
45						
46						
47	Lanzhou Large-tailed sheep	LLT	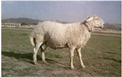	Domestic_East Asia_Mongolia	2	White wool
48						

**Table 2 animals-13-03265-t002:** The distribution of SNP variants in the genome region.

Catalogue	SNP Numbers
Upstream	93,281
Exonic	151,411
Intronic	8,123,336
Splicing	4227
Downstream	125,137
upstream/downstream	2550
Intergenic	13,414,800
ts	14,501,815
tv	7,631,392
ts/tv	1.9
Total	22,133,207

**Table 3 animals-13-03265-t003:** Genes associated with wool color inferred from Sankey diagrams.

Category	Gene	Number of Relevant Pathways
Non-White	*PPP1CB*	7
	*CALML4*	9
	*PPP3CA*	10
	*GRM5*	6
	*GRIN1*	8
	*MYLK*	6
	*FGF18*	5
	*FGFR2*	6
	*FGF2*	5
	*MAPK10*	8
	*NOS3*	6
White	*ACAT2*	5
	*PCCB*	2
	*ALDH6A1*	2
	*ACSS2*	2
	*PAPSS2*	2

## Data Availability

The data are available upon request due to privacy/ethical restrictions.

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
