# Peer review of "Whole Genome Resequencing Reveals Selection Signals Related to Wool Color in Sheep"

_animals, 2023, doi:10.3390/ani13203265_

Round 1
Reviewer 1 Report
General Comments
This research holds significant importance by providing crucial insights into the genetics of wool color in sheep. This knowledge enables selective breeding for desirable wool colors, improving wool quality, diversifying textile options, and stimulating economic growth in the wool industry. Additionally, it contributes to our broader understanding of livestock genetics and offers practical applications for breeding strategies, ultimately benefiting both the textile sector and genetic research. The analysis is well-executed, utilizing a robust dataset, and the study design is commendable, providing valuable insights. However, there are certain areas that should be addressed to enhance the manuscript's potential for publication.
Specific comments:
Point 1: In Sample Collection and Sequencing (2.2), could you please specify the year of the experiment you conducted?
Point 2: In Table 1, please specify the complete name of the Category column abbreviation in the table's legend.
Point 3: In Table 3, please format the gene name in italics.
Point 4: Please review and ensure proper spacing is applied before references throughout the entire manuscript text.
Author Response
Dear Reviewer:
Greetings! I would like to express my sincerest gratitude to you, it is an honor for me to be reviewed by you. I would like to thank you for taking your time to scrutinize our manuscript and for your valuable suggestions. For each of your suggestions, I have made detailed and comprehensive revisions, which are specified in the attached document.
Sincerely
Caihong Wei

Reviewer 2 Report
The manuscript aimed to identify pathways and genes associated with wool color in sheep using whole genome resequencing in different breeds. The search for genes that influence coat color is an interesting area that, in the case of wool, is of economic and breeding interest, but also provides insight into the genetic background of pigmentation.
General comments
When evaluating the results, two aspects should be taken into account. i) How is white wool basically caused in sheep? Are white sheep albinos or leucists? Answering this question would significantly reduce the number of candidate genes for different color. ii) The sheep included in the study show different wool colors, but at the same time belong to different breeds (see 3.1). There are no sheep that belong to the same breed but have different wool color. This is not necessarily a problem since several breeds are grouped together in the different color groups. But it must be noted when discussing candidate genes.
Additional comments
1 The introduction is long and does not only refer to the question posed.
2 There is no clear demarcation between the color groups in the representations. Table 1: Is “Shipping Gray” white or non-white? Table 3: Where is the boundary between the genes?
3 The population analysis (3.1) is interesting, but not the aim of the study. The statements about the endangered races in the discussion also do not fit properly into the task. In addition, only the most important part of the five partial images in Figure 1 should be shown. The rest could go into the supplementary material.
4 There are several very general references to previous studies (lane 312, 320). If it is necessary to refer to previous studies, these must also be cited.
5 It is understandably difficult to process the large number of significant GO terms and pathways, but it is necessary to justify the reduction. The information “etc.” (e. g. lane 316, 324, 337, 551 and also in the abstract) does not work at all.
6 Figure 3: The names of genes and pathways are very difficult to read.
7 Table 3 does not follow from the previous results presentations.
8 4.1 mostly does not match the title and the task.
9 Lane 371: three populations, not four?
10 Lane 391: The right term is leucism. Make a clear distinction between albinism and leucism (Lane390-398)
11 Part 4.2.1 is primarily a list of pathways that may be related to pigment formation. There is no rating here, although it is part of the discussion. GO terms are named that were never mentioned before. Therefore, it is not clear which own results agree with the literature. Or were all of these GOs also found in the own study? This does not take into account the fact that certain pathways can also be influenced by the different breeds with their different wool properties (wool diameter, wool distribution, wool length, etc.) and generally diverse phenotypes.
12 Some genes are important for all cell activity that their specific influence on wool color is rather marginal (e.g. GATA1, MAPK, IGF1R)
13 Conclusions: Should be revised and thus present your own conclusions more clearly. Some facts have long been known (Lane 549, 557-558).
Author Response

(The authors gave the same response as above.)

Round 2
Reviewer 2 Report
No further comments.